# Stable representation of sounds in the posterior striatum during flexible auditory decisions

Lan Guo[1], William I. Walker[1], Nicholas D. Ponvert[1], Phoebe L. Penix[1] & Santiago Jaramillo [1]

The neuronal pathways that link sounds to rewarded actions remain elusive. For instance, it is unclear whether neurons in the posterior tail of the dorsal striatum (which receive direct input from the auditory system) mediate action selection, as other striatal circuits do. Here, we examine the role of posterior striatal neurons in auditory decisions in mice. We find that, in contrast to the anterior dorsal striatum, activation of the posterior striatum does not elicit systematic movement. However, activation of posterior striatal neurons during sound presentation in an auditory discrimination task biases the animals' choices, and transient inactivation of these neurons largely impairs sound discrimination. Moreover, the activity of these neurons during sound presentation reliably encodes stimulus features, but is only minimally influenced by the animals' choices. Our results suggest that posterior striatal neurons play an essential role in auditory decisions, and provides a stable representation of sounds during auditory tasks.

[1] Institute of Neuroscience, Department of Biology, University of Oregon, Eugene, OR 97403, USA. These authors contributed equally: Lan Guo, William I. Walker. Correspondence and requests for materials should be addressed to S.J. (email: sjara@uoregon.edu)

In the mammalian brain, the dorsal striatum links neural signals from the cerebral cortex to circuits in the basal ganglia to mediate action selection. Electrophysiological and inactivation studies have identified two regions within the dorsal striatum which play distinct roles in decision making: the dorsomedial striatum (DMS) involved in flexible goal-oriented behavior, and the dorsolateral striatum (DLS) which mediates habitual actions[1–3]. Recent anatomical characterization of the excitatory input from cortex and thalamus onto the striatum suggests that the organization of the dorsal striatum goes beyond the DMS and DLS divide[4]. This characterization in rodents showed that the posterior portion of the striatum receives a combination of sensory inputs that sets it apart from other regions. Similarly, an evaluation of reward-related signals of the dopaminergic input along the anterior–posterior axis of the striatum provides further evidence that the posterior tail of the striatum forms a circuit distinct from the anterior dorsal striatum, which includes the classically studied DMS and DLS regions[5]. It is not clear, however, whether the function of this posterior region is qualitatively different from the previously characterized striatal subregions. Here, we evaluate the role of neurons in the posterior tail of the striatum during sensory-driven decisions in mice.

In primates, neurons in the tail of the caudate nucleus (part of the dorsal striatum) respond to visual stimuli[6] and encode stimulus value[7]. Moreover, neurons in the primate caudate causally contribute to visual perceptual decisions[8]. In contrast, little is known about the role of dorsal striatal neurons during auditory decisions tasks. The posterior tail of the dorsal striatum in rodents (referred to hereafter as posterior striatum) receives direct neuronal projections from the auditory thalamus (ATh) and the auditory cortex (AC), as well as midbrain dopaminergic signals[4,9]. Because of these anatomical features, this region is sometimes referred to as the auditory striatum[10]. Given this convergence of sensory and reward-related signals, and prompted by the role of other dorsal striatal regions, we hypothesized that the posterior striatum drives rewarded actions according to acoustic cues. Here, we show that such a hypothesis does not fully account for the role of this striatal region during sound-driven decisions. Our findings show that posterior striatal neurons are necessary for the expression of sound-action associations, and that stimulation of these neurons biases decisions based on sounds. In contrast to activation of anterior dorsal striatal neurons, activation of posterior striatal neurons does not promote movement outside of sound discrimination tasks. Moreover, when a behavioral task requires rapid updating of sound-action associations without changes in the expected reward, the representation of sounds by the large majority of posterior striatal neurons is stable across contexts and does not depend on the animal's choice. These results suggest that once an animal has learned a sound-driven decision task, neurons in the posterior striatum provide sensory information downstream, while providing little information about behavioral choice before action initiation.

## Results

**Posterior striatum does not promote movement outside a task.** The striatum is comprised of two main neuronal outputs, the direct (or striatonigral) pathway and the indirect (or striato-pallidal) pathway. One experimentally supported model of dorsal striatal function posits that the striatal direct pathway promotes action initiation[11,12]. To test whether activation of the posterior striatum produces similar effects on motor initiation as the anterior dorsal striatum (referred to hereafter as anterior striatum), we used *Drd1a::ChR2* mice which express channelrhodopsin-2 (ChR2) in direct-pathway medium spiny neurons (dMSNs), and optogenetically activated these neurons in freely moving animals (Fig. 1a).

We found that unilateral optogenetic stimulation of anterior dorsal striatal dMSNs in freely moving mice elicited contralateral head rotation often followed by whole body rotation toward the contralateral side clearly visible in single trials (Fig. 1b, green trace). In contrast, unilateral stimulation of the same magnitude and duration in the posterior striatum did not result in head or body rotation (Fig. 1b, purple trace). Analysis of each animal showed a clear difference between stimulating the left vs. right hemisphere in the anterior striatum ($p < 0.05$ for each of 4 mice, Wilcoxon rank-sum test), but no significant effect in the posterior striatum ($p > 0.1$ for each of 3 mice, Wilcoxon rank-sum test). Across the mice tested, the average head rotation to the contralateral side of stimulation was $121 \pm 45°$ (mean ± SD) for anterior striatum and $4 \pm 10°$ for posterior striatum. This difference was statistically significant on each hemisphere (Fig. 1c, left hemi: $p = 0.034$; right hemi: $p = 0.034$, Wilcoxon rank-sum test). This difference cannot be explained by a difference in the density of dMSNs between the anterior and the posterior striatum. Characterization of the striatum in the *Drd1-Cre* mice (founder line FK150) yielded similar numbers of *Drd1*-positive neurons ($1654 \pm 92$ cells per square millimeter per section in the anterior striatum vs. $1709 \pm 127$ cells in the posterior striatum; $p = 0.529$, Wilcoxon rank-sum test), consistent with reports in other *Drd1-Cre* lines[13,14].

These results indicate that activation of direct-pathway neurons in the posterior striatum outside a behavioral task does not directly promote movement, in contrast to other dorsal striatal regions. Because the posterior striatal neurons receive dense auditory input, we set out to evaluate the function of these neurons in the context of sound-driven decisions.

**Posterior striatal dMSN activation biases auditory decisions.** To test the role of posterior striatal neurons in sound-driven decisions, we first evaluated the effects of activating posterior striatal dMSNs during a two-alternative choice sound discrimination task. Mice initiated a trial by poking their nose in the center port of a 3-port chamber. After the presentation of a 100 ms sound, mice were required to go to either the left or right reward port based on the frequency of the acoustic stimulus (Fig. 2a). To test the effect of activating posterior striatal dMSNs in the task, the same group of mice in which optogenetic stimulation did not elicit rotational movement (Fig. 1) were tested while they performed the task. Optogenetic stimulation for 200 ms (50 msec before sound onset to 50 msec after sound offset) was presented in 20% of the trials. Unilateral activation of posterior striatal dMSNs during sound presentation resulted in a contralateral choice bias apparent in single sessions (Fig. 2b, c). Average performance was clearly different between trials with stimulation of the left hemisphere and stimulation of the right hemisphere ($p < 0.001$, Wilcoxon rank-sum test, Fig. 2d). Significant differences in bias were observed between left hemisphere and right hemisphere stimulations for each *Drd1a::ChR2* mouse tested ($p < 0.01$ for each of 3 mice, Wilcoxon rank-sum test), although smaller effects were obtained when the location of the optical fiber (confirmed postmortem) was close to the border between the striatum and the cortex (Supplementary Fig. 1). No effect was observed when the optogeneitc stimulation was applied to wild-type mice, confirming that the observed bias was not a result of visible light stimulation (Supplementary Fig. 1).

In addition, we tested whether the bias direction depended on the multiunit frequency tuning of the stimulated site. We measured the multiunit sound response of each stimulated site through a tetrode bundle implanted alongside the optical fiber.

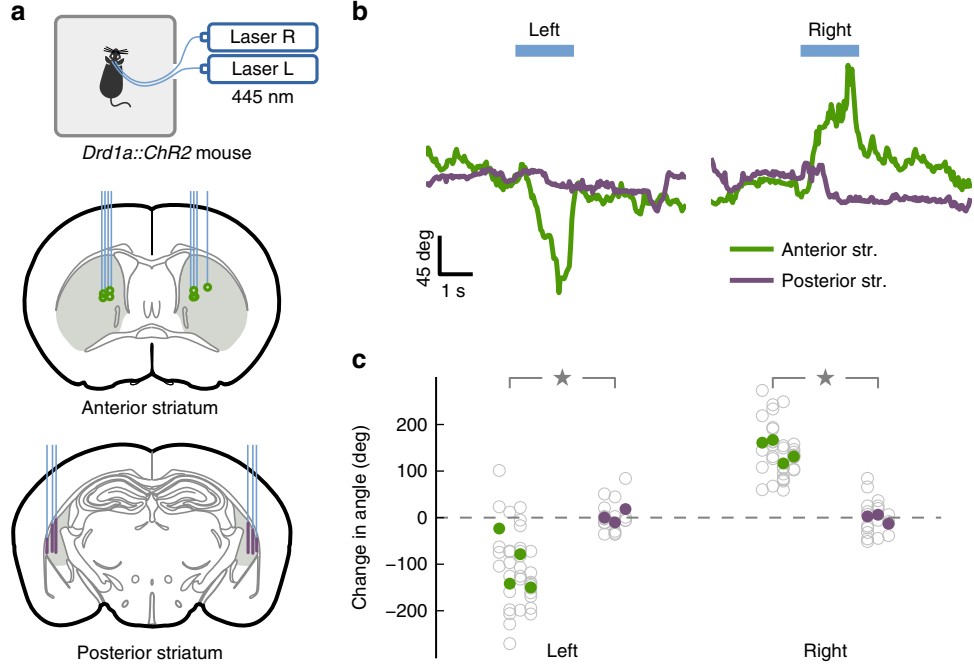

**Fig. 1** Activation of distinct subregions of the dorsal striatum produced different effects on movement. **a** Top: experimental setup. Optogenetic stimulation in freely moving mice of direct-pathway neurons from one of four different sites in the dorsal striatum: anterior striatum (left or right) and posterior striatum (left or right). Middle: Coronal brain slice. Green dots indicate the tip of fixed optical fibers implanted in the anterior striatum (gray) confirmed postmortem. Bottom: purple lines indicate the stimulation sites by movable optical fibers implanted in the posterior striatum. **b** Representative head angle trace over one trial of unilateral stimulation at each site. The blue bar represents the laser pulse (1.5 s) delivered in each trial. Positive angles correspond to left rotation. **c** Average change in head angle by optogenetic stimulation in each mouse tested. Each gray circle is one trial, each filled circle is the average for one hemisphere of one mouse. Stimulation of anterior striatum (green) produced a significantly larger change in head angle compared to stimulation of the posterior striatum (purple) in either hemisphere ($p = 0.034$, Wilcoxon rank-sum test)

Both hemispheres contained sites responsive to sound frequencies above or below the sound categorization boundary. We found no significant correlation between the preferred frequency of a stimulated site and the direction of the observed behavioral bias ($r = 0.135$, $p = 0.478$, Spearman correlation test, Supplementary Fig. 2). A contralateral bias without clear frequency-specific bias is reminiscent of the effects observed when stimulating the AC without cell-type specificity (Supplementary Material of ref. [10]).

These results indicate that activation of posterior striatal neurons during the sound presentation biases decisions in the auditory task, even though similar stimulation does not promote movement outside the task. We next set out to test the necessity of these neurons for sound-driven decisions.

**Posterior striatum inactivation impairs auditory decisions**. To test whether the activity of posterior striatal neurons was required when performing the sound-discrimination task, we quantified task performance during bilateral reversible inactivation of these neurons (Fig. 3, Supplementary Fig. 3). Injection of muscimol (a GABA-A receptor agonist) in the posterior striatum resulted in a consistent decrease in task performance compared to injection of saline as control (Fig. 3b). The effect was observed in all mice tested (Fig. 3c, $p < 0.05$ for each mouse, Wilcoxon rank-sum test), and consisted of a flattening of the psychometric curve. Repeated injections did not alter task performance across the saline sessions tested ($r = -0.24$, $p = 0.75$, linear regression of accuracy on sessions). These results were replicated using fluorescent muscimol to confirm that inactivation restricted to the posterior striatum affected task performance (Supplementary Fig. 4). As a result of muscimol inactivation, mice displayed slower withdrawals from the center port on average ($p < 0.05$ for four out of five mice, Wilcoxon rank-sum test, Supplementary Fig. 5). A change in

movement speed from the center port to the reward ports was observed in only one out of five mice ($p < 0.05$, Wilcoxon rank-sum test, Supplementary Fig. 5). Even though animals displayed motor impairments during muscimol inactivation, they still performed hundreds of trials during each session ($480 \pm 146$ muscimol vs. $733 \pm 86$ saline, Supplementary Fig. 5, $p < 0.05$ for three out of five mice, Wilcoxon rank-sum test). During inactivation sessions, some animals used a strategy in which they chose a reward port at random, while other animals displayed a strong bias to one side. Both of these strategies resulted in an average performance close to chance level for the binary choice.

These inactivation results indicate that the activity of neurons in the posterior striatum is necessary for auditory decisions. We next quantified what information is encoded by posterior striatal neurons during sound-driven decisions.

**Posterior striatal neurons encode sounds and actions**. To further delineate the possible functional roles of posterior striatal neurons, we recorded activity from single neurons and examined their response to different components of the sound-driven decision task. We first evaluated the neuronal responses to sound stimuli presented in the two-alternative choice task.

We found clear neural responses to sounds and selectivity to sound frequency in neurons from the posterior striatum. Figure 4a and b shows sound responses from two distinct neurons during the discrimination task. To estimate how many neurons were responsive to sounds during the task, we compared the firing rate of each neuron during the sound presentation (all trials pooled together) against the neurons spontaneous firing (Fig. 4c). We found that 44.6% (232/520) of cells showed a significant change in firing in response to sound ($p < 0.05$, Wilcoxon rank-sum test). From the sound responsive neurons, 48.7% (113/232) responded

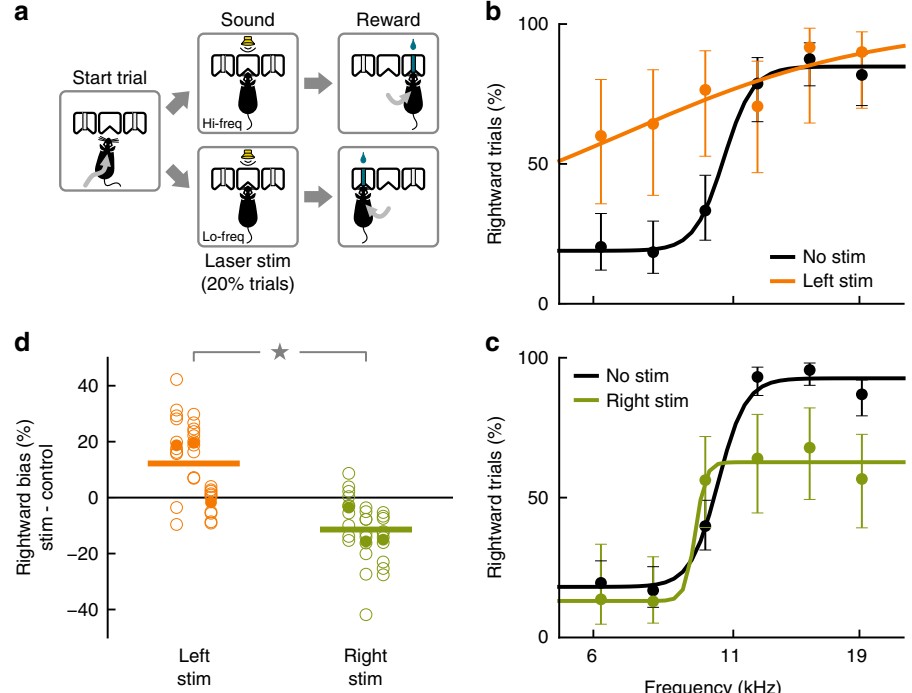

**Fig. 2** Activation of direct-pathway posterior striatal neurons biased sound-driven decisions. **a** Schematic of the two-alternative choice sound frequency discrimination task. Mice initiated each trial by entering a center port and had to choose one of two side reward ports depending on the sound presented: low-frequency = left, high-frequency = right. **b** Psychometric performance for one behavioral session that included optogenetic activation of direct-pathway neurons in the left posterior striatum on 20% of trials. Error bars indicate 95% confidence intervals. **c** Psychometric performance for one behavioral session that included optogenetic activation of direct-pathway neurons in the right posterior striatum. **d** Change in the percentage of rightward choices during optogenetic stimulation for each hemisphere in each mouse tested. Each open dot represents one session, each filled dot represents the average bias for that hemisphere in one mouse (N = 3 mice, 10 sessions each hemisphere per mouse). Horizontal bars represent averages across all sessions for all mice. Stimulation produced significantly different biases in the left vs. the right hemisphere (p < 0.001, Wilcoxon rank-sum test)

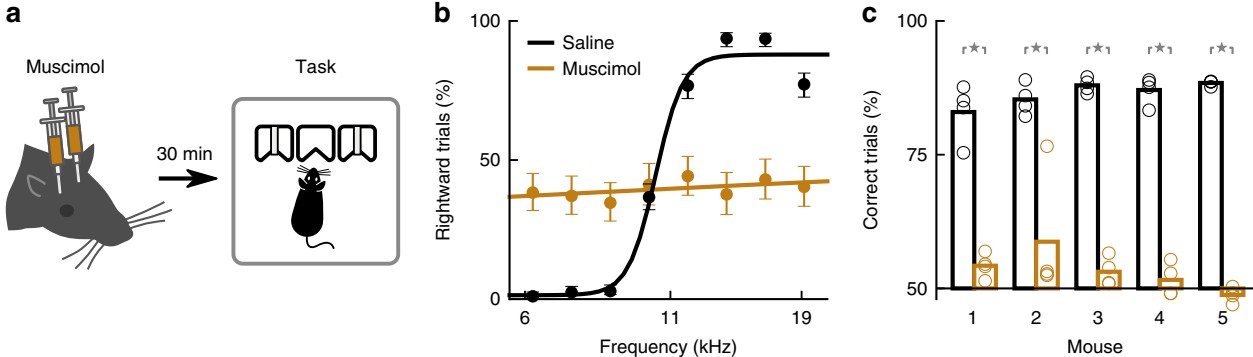

**Fig. 3** Inactivation of posterior striatal neurons impaired sound-driven decisions. **a** Mice were injected bilaterally with muscimol in the posterior striatum 30 min before they performed the two-alternative choice sound discrimination task. **b** Average psychometric performance for one mouse on sessions with injection of muscimol (four sessions) or saline control (four sessions). Error bars indicate 95% confidence intervals. **c** Average percentage of correct trials on each saline session (black) and each muscimol session (brown) for each mouse. Bars indicate average across sessions for each mouse. Muscimol inactivation significantly reduced the percentage of correct trials on each mouse (p = 0.021, Wilcoxon rank-sum test)

differently across sound stimuli of different frequencies (p < 0.05, Kruskal–Wallis H-test). We further quantified how many neurons provided sound information relevant to the task by comparing sound-evoked responses between high-frequency (any sound above the categorization boundary) vs. low-frequency trials (any sound below the boundary). Because sound-responsive neurons displayed preferred stimuli across the frequency spectrum used in the task, we expect these neurons (with the exception of those that are broadly tuned or tuned to a frequency at the categorization boundary) to contribute to the perceptual decision. Consistent with this, we found that 26.3% (137/520) of all recorded cells, that is 40.9% (95/232) of sound-responsive cells, showed a different average response between high-frequency and low-frequency sounds during the task (p < 0.05, Wilcoxon rank-sum test, Fig. 4d). Sound responses and frequency selectivity were also observed when sounds were presented outside the context of the discrimination task (Supplementary Fig. 6). These measurements indicate that the identity of sounds has a strong influence on the firing rate of neurons in the posterior striatum.

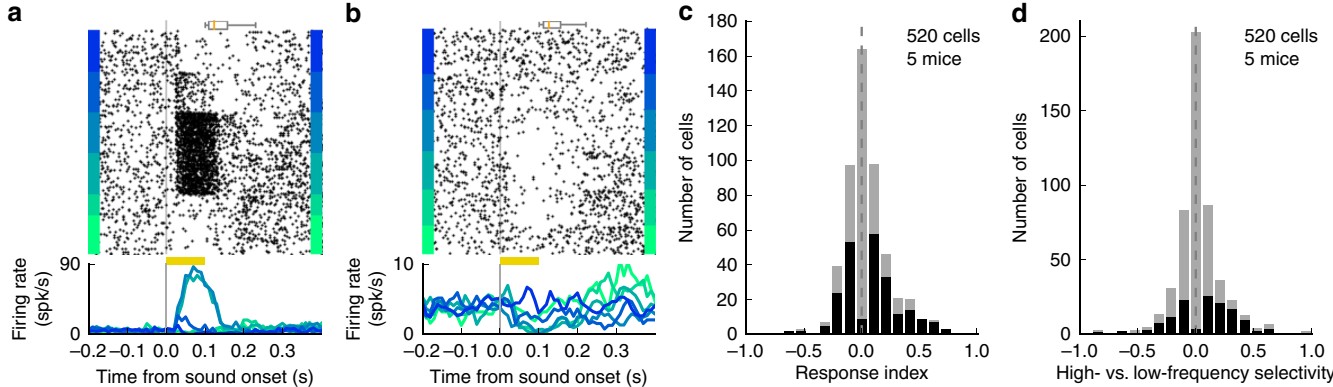

**Fig. 4** Posterior striatal neurons displayed frequency-selective sound-evoked responses. **a** Example of sound responses from a posterior striatal neuron during the sound discrimination task. Yellow bar indicates the duration of the sound (100 ms). A box plot above the spike raster shows the distribution of center-port exit times. **b** Sound responses of a different posterior striatal neuron which showed only suppression of activity. Both example cells showed clear frequency selectivity. **c** Magnitude of sound-evoked response estimated from the 100 ms period during sound presentation for each neuron (all trials pooled together). A positive response index indicates an increase in activity compared to baseline spontaneous activity. A negative index, a decrease in activity. Neurons with statistically significant evoked responses ($p < 0.05$, Wilcoxon rank-sum test) are shown in black. **d** Sound selectivity index calculated by comparing neural responses to high- vs. low-frequency sounds based on the categorization boundary from the task. Neurons with statistically significant differences ($p < 0.05$, Wilcoxon rank-sum test) are shown in black

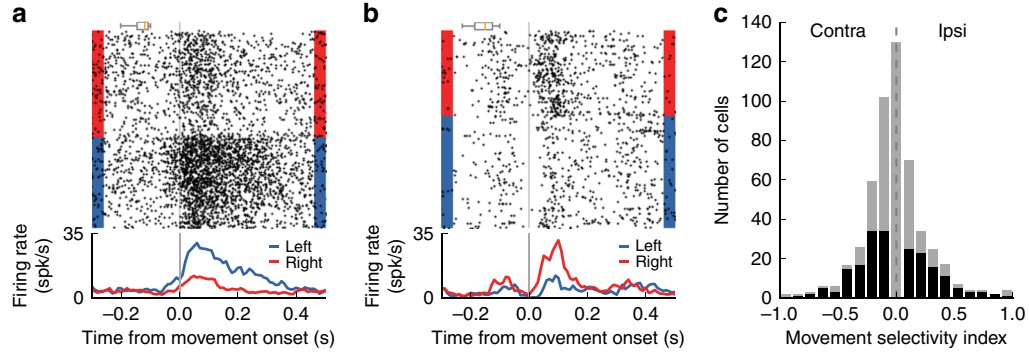

**Fig. 5** Firing rate of posterior striatal neurons during movement depends on movement direction. **a** Activity from a posterior striatal neurons aligned to the moment the mouse left the center port and moved toward a reward port. Activity differed between right and left choices. Plot includes all trials (any stimulus frequency). **b** Activity from a different posterior striatal neuron showing the opposite movement selectivity. Box plots above spike rasters in **a** and **b** show the range of onset times for the sound stimuli. **c** Movement selectivity index: $(I - C)/(I + C)$, where $I$ and $C$ are the average firing rates in the period 50–150 ms after leaving the center port for choices ipsilateral and contralateral to the recording hemisphere, respectively. $N = 520$ cells recorded from five mice. Cells with a statistically significant difference in activity during contralateral and ipsilateral choices ($p < 0.05$, Wilcoxon rank-sum test) are shown in black

To examine whether the activity of posterior striatal neurons was correlated with the movement of the animals, we quantified the difference in firing rate during movement toward the left port vs. the right port. After an animal left the center port, the activity of these neurons was often different depending on movement direction (Fig. 5a, b). We found that this effect was significant ($p < 0.05$, Wilcoxon rank-sum test) for 38.5% (200/520) of all cells recorded (Fig. 5c). Of the subset of cells that were selective to movement direction, 55.5% showed stronger firing during movement contralateral to the recording hemisphere. These recordings include MSNs from both the direct and indirect pathways, thus the equivalent proportions of cells showing stronger activity during ipsi vs. contra movement is not necessarily in contradiction to the bias observed when activating only dMSNs during sound presentation in the task. Across the whole population, more cells fired at higher rates during movement contralateral to the recording site ($p = 0.025$, Wilcoxon signed-rank test). Last, we evaluated the relation between left/right movement selectivity and low/high sound frequency selectivity. We found that 11.9% (62/520) of neurons were both

selective to sound frequency and movement direction during the behavioral task (Supplementary Fig. 7). For these 62 cells, the frequency selectivity index was positively correlated with the movement selectivity index ($r = 0.606$, $p < 0.001$, Spearman correlation test), indicating that cells that responded more strongly to low-frequency sounds also displayed more activity during movement to the left reward port.

These results indicate that at different time periods during the behavioral task, posterior striatal neurons encode information about sounds and actions. However, these observations are not sufficient to know whether these neurons encoded the sensory-motor association during the sound presentation, or simply relay auditory information before movement initiation. We therefore tested whether the neuronal activity during sound presentation was influenced by the animal's choice.

**Effects of choice on posterior striatum sound responses.** One possible model for auditory decision making assumes that the posterior striatum is a main locus of action selection, given

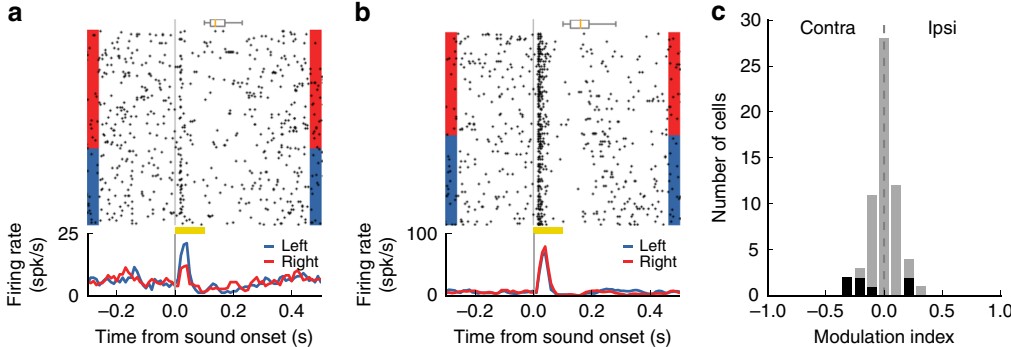

**Fig. 6** Choice influenced sound-evoked responses in a subset of posterior striatal neurons. **a** Response of one neuron to a stimulus near the categorization boundary, grouped according to the animal's choice. Sound-evoked response for this neuron was modulated by the animal's choice. The box plot above the spike raster shows the distribution of center-port exit times. **b** Activity of a different neuron showing no influence of choice on the sound-evoked response. **c** Influence of choice on sound-evoked activity (from the 100 ms period during sound) for all neurons that showed a response to stimuli near the categorization boundary ($N = 67$ cells from five mice). Less than 12% of neurons showed a significant modulation by choice ($p < 0.05$, Wilcoxon rank-sum test), shown in black

sensory information from cortex and thalamus. This model predicts that sound-evoked responses of a striatal neuron will differ depending on the animal's action, even if the stimulus is the same. To test this prediction, we compared neural activity evoked by the same sound between trials in which a mouse chose the left reward port and trials in which the mouse chose the right reward port.

We focused our analysis on trials with sound stimuli near the categorization boundary for which animals performed near 50% accuracy (i.e., the numbers of trials with a left or right choice were comparable). Figure 6a shows a posterior striatal neuron in which sound-evoked responses were influenced by the animal's choice. However, for most neurons, the animal's choice had little to no influence on the evoked responses, as in the example neuron shown in Fig. 6b. From our analysis of posterior striatal neurons that responded to stimuli near the categorization boundary, we found that 11.5% (7/61) had a significant change in evoked response depending on the animal's choice (Fig. 6c, $p < 0.05$, Wilcoxon rank-sum test). Different striatal cell types display different spike shapes[15]; however, analysis showed no systematic relation between the spike shape of each neuron and the magnitude of its response modulation by choice (Supplementary Fig. 8). Moreover, the magnitude of modulation by choice was not correlated to the magnitude of movement selectivity in these cells (Supplementary Fig. 9, $r = 0.02$, $p = 0.8$, Spearman correlation test).

To test whether posterior striatal neurons encoded the animal's choice while in the center port, we also quantified the activity of each neuron during the 100 ms window before mice left the center port. This analysis included only trials with the stimulus near the boundary, to control for the changes in activity due to sound frequency. We found that 4.6% (24/520) of neurons displayed significantly different activity before action initiation based on the subsequent choice (Supplementary Fig. 10, $p < 0.05$, Wilcoxon rank-sum test). This fraction was lower than the proportion of cells that encoded task-relevant sound information (high- vs. low- frequency), even when we factored out the influence of subsequent choice and matched the number of trials between these two analyses (4.6% for choice vs. 8.2% for sound, $p < 0.05$, Fishers exact test).

Overall, these results suggest that in the case of ambiguous (difficult) stimuli, activity of posterior striatal neurons before action initiation contains little information about the animal's subsequent choice. However, these results do not rule out the possibility that information about choice would be present in

sound-evoked responses when an animal has formed a clear association between a stimulus and an action.

**Posterior striatum activity during flexible categorization.** In the experiment described in the previous section, the animals' variability in choice arose from the difficulty of perceptual decisions near the decision boundary. To directly test whether neuronal activity in this region encodes stimulus-action associations, we performed additional recordings using a task in which these associations are reversed. We hypothesized that a larger number of neurons would be influenced by the animal's choice if instead of using ambiguous stimuli, the meaning of the stimuli changed systematically from predicting reward on the left port to predicting reward on the right port.

To test this hypothesis, we used a previously developed task for rodents[16] in which the rewarded action associated with a subset of sounds changes every few hundred trials (Fig. 7a). In this switching task, low-frequency sounds still indicate reward on the left side (and high-frequency indicates reward on the right), but the categorization boundary changes across blocks of trials. As a result, a stimulus of intermediate frequency (e.g., 11 kHz) is associated with the left port in one block of trials and associated with the right port on the next block. Importantly, the amount of reward associated with such sound remains the same, and it is only the action associated with this sound that changes. Mice trained in this flexible categorization task achieved high-performance levels and were able to switch between sound-action association contingencies several times per session (Fig. 7b). Consistent with previous reports[16], well-trained mice took less than 20 trials to establish the new stimulus-action association (Supplementary Fig. 11).

We found that, in a subset of posterior striatal cells, the evoked response to a given sound changed depending on the port associated with that sound (Fig. 7d). However, most cells showed a stable representation of sounds across blocks of trials, independent of the associated reward port (Fig. 7e). From our analysis of neurons that responded to the stimulus of intermediate frequency, we found that sound-evoked responses in 12.9% (20/155) of cells were influenced by changes in the sound-action association (Fig. 7c, $p < 0.05$, Wilcoxon rank-sum test). Similarly, when we evaluated neuronal activity before mice left the center port in these trials, 7% (51/725) of cells showed a statistically significant change in activity according to the animal's choice (Supplementary Fig. 10).

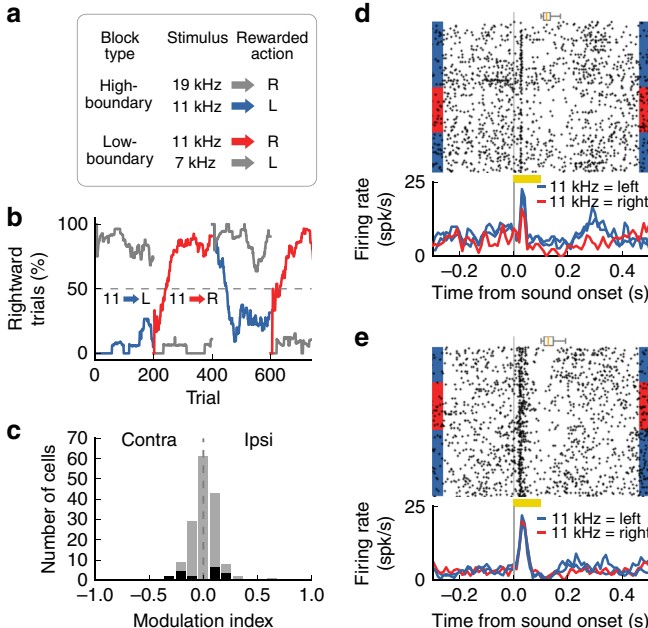

**Fig. 7** Effect of rapid changes in sound-action associations on activity of posterior striatal neurons. **a** Switching task: the rewarded action associated with a sound of intermediate frequency changed from one block of trials to the next. **b** Example performance of one mouse (one session) as the contingency changes. Colors match those in **a**. **c** Influence of changing sound-action association on sound-evoked activity (in the 100 ms period during sound) for all neurons that showed a response to the switching stimulus ($N = 155$ cells from four mice). Less than 13% of neurons showed a significant change across sound-action contingencies ($p < 0.05$, Wilcoxon rank-sum test), shown in black. **d** Responses of one posterior striatal neuron to the stimulus of intermediate frequency for three blocks of trials. Only correct trials are included. Sound-evoked responses for this neuron changed systematically depending on the rewarded action associated with the stimulus. **e** Activity of a different neuron showing no change in sound-evoked responses across sound-action contingencies. Box plots above spike rasters show the distribution of center-port exit times

These results provide further support for the idea that posterior striatal neurons display a stable representation of sounds, minimally affected by action selection. Under the conditions tested, the activity of posterior striatal neurons encoded information about sound identity, rather than the behavioral choice or the sound-action association.

## Discussion

The main objective of this study was to evaluate the role of neurons in the posterior tail of the rodent striatum during sensory-driven decisions. Our experiments helped evaluate which roles these neurons play in the pathway linking sensation to action: from representing the raw stimulus, to performing computations necessary for stimulus discrimination, choice selection, or action execution. Based on previous studies of the dorsal striatum[11,12,17], together with evidence of synaptic plasticity in the cortico-striatal pathway during reward-driven learning[18,19], we posited that posterior striatal neurons would promote actions according to rewarded associations to acoustic stimuli. We found instead, that after animals have learned a sound discrimination task, the activity of these neurons before action initiation contained reliable information about the identity of the stimulus and that this neuronal activity was not considerably affected by the animal's subsequent choice.

In contrast to the fine parcellation commonly used to describe the cerebral cortex, a functional subdivision of the dorsal striatum is usually limited to a lateral region (involved in habitual behaviors) and a medial region (involved in flexible behaviors). However, studies in mouse and rat evaluating the inputs onto striatal circuits from cortical, thalamic, and dopaminergic neurons demonstrate a more refined anatomical organization of the dorsal striatum which in turn suggests the existence of further functional specialization[4,5,20,21]. A characterization of the distribution of dopamine receptors across the rostro–caudal axis provides further support for this refinement[13]. In particular, the posterior portion of the rodent striatum has been identified as an area that receives convergent inputs from sensory cortex, sensory thalamus, and midbrain dopamine neurons.

This functional specialization is reflected in the different effects observed when activating different subregions of the dorsal striatum. In contrast to the results of activating neurons in the anterior dorsal striatum, optogenetic activation of direct-pathway neurons in the posterior striatum did not elicit movement outside the behavior task. This result indicates that these cells play a distinct role in motor initiation from neurons in the anterior portion. It is also possible that posterior striatal neurons promote specific actions or choices, but they are only recruited during behavioral tasks where auditory stimuli are present and associated with specific actions for reward. The additive perceptual bias observed when activating striatal neurons during visual detection tasks[22] provides support for this hypothesis.

In an auditory decision task, activation of rat auditory cortico-striatal neurons produced a choice bias that depended on the frequency tuning of the stimulated site[10]. In the same study, however, optogenetic stimulation of auditory cortical neurons without cell-type specificity caused a contralateral choice bias that did not depend on frequency tuning (see Supplementary Fig. S3 of their study[10]). Our results when stimulating dMSNs in the posterior striatum during sound discrimination recapitulated the latter finding by producing a consistent contralateral bias with only a weak (and not statistically significant) correlation to stimulation site tuning. Given that both direct-pathway and indirect-pathway MSNs receive input from cortico-striatal neurons, our results do not rule out the possibility that a balance between dMSN and iMSN activity in the posterior striatum is required to produce a frequency-specific bias. Moreover, it is possible that during learning, posterior striatal neurons that represent sounds rewarded on the contralateral port potentiate their connections to downstream neurons that promote contralateral movement. In this scenario, our data is consistent with posterior striatal neurons conveying mostly information about the identify of the stimulus which is disturbed during optogenetic stimulation, causing the erroneous decisions seen in Fig. 2d.

Lesion studies indicate that rodents can perform sound-action association tasks without the AC[23–26]. In contrast, lesions of the ATh largely affect expression of fear conditioning to sounds[24] and discrimination of sound frequencies[26]. These observations suggest that outputs of the ATh convey signals essential for auditory decisions. The dorsal striatum, a direct target of the ATh with connections to motor structures, is therefore a likely candidate circuit to mediate tasks that require sound-driven decisions.

Our inactivation experiments demonstrate that silencing auditory striatal neurons has a drastic effect on a task that requires sound-driven decisions. One interpretation of these results states that these neurons belong to a unique pathway required for successful performance of the frequency discrimination task we study. Alternatively, these neurons could form one of several parallel pathways linking sensation to action, and reversible inactivation yields strong effects on behavior by

altering the dynamics of downstream circuits[27]. Our experiments cannot distinguish between these possibilities, yet, provide strong support for a role of these striatal neurons in sensory-driven decisions.

When examining activity in this region of the striatum during auditory task in mice, our results demonstrate that a large fraction of neurons have clear selectivity to sound features, consistent with previous observations in the rat[10,28]. In our sound discrimination task, over a quarter of the recorded posterior striatal neurons were able to distinguish between sound frequencies above and below the categorization boundary, and a larger percentage discriminated across different stimuli (even within a category). These findings suggest that the fine sound frequency selectivity in posterior striatal neurons does not result simply from associating a sound to left vs. right actions. In addition, when tested with stimuli close to the boundary, we found little information about the animals' choices encoded in the neural activity during the presentation of the sound. Although our task does not allow for an accurate estimate of the moment the decision is made, differences in neural activity before action initiation would be expected if a neuron encodes stimulus-action associations. However, the majority of neurons did not show such differences. We conclude therefore that few sound-responsive neurons in this brain region (less than 13% in this study) encode the sound-action association on each trial. This result is comparable to observations from the ATh and AC of rats during a similar task[29], which raises the possibility that activity changes observed in the striatum simply reflect modulated signals arriving from thalamic or cortical inputs. Taken together, these results support a model in which posterior striatal neurons provide sensory information rather than encoding the behavioral choice in well-learned tasks.

All measurements in our study were performed after animals had achieved high-performance in the discrimination task. Therefore, our data does not provide evidence for the role of the posterior striatal circuits during learning. A recent study found that auditory cortico-striatal synapses were strengthened as rats learned to perform a sound discrimination task[19]. Together with our results, these findings suggest that auditory cortico-striatal circuits undergo major changes in synaptic strength during the learning of sound-action association tasks, but do not display such changes when a well-learned task requires rapid switching in the associations between sounds and actions without changes in reward (Fig. 7). We conclude that after establishing the initial associations, posterior striatal circuits convey sensory information that can be rerouted downstream to drive different responses to the same sensory stimulus.

The conclusion above is consistent with a model in which posterior striatal neurons encode the value of sensory stimuli, such that evoked responses vary significantly when a stimulus becomes predictive of reward (when learning the task for the first time), but do not change when the stimulus predicts the same reward under different stimulus-action contingencies. This is reminiscent of the coding for action-value observed in anterior striatal neurons[30]. Coding of stimulus-value by posterior striatal neurons would result in a stable representation of sound features when the association between these stimuli and rewards remains constant, even when animals must modify their behavioral responses in order to obtain reward.

## Methods

**Animal subjects.** Seventeen adult male wild-type mice (C57/BL6J) and 7 transgenic adult male mice were used in this study. Transgenic mice expressing *Cre* recombinase under control of the dopamine D1 receptor (036916-UCD from MMRRC) were crossed with *LSL-ChR2* mice (012569 from JAX) to produce mice expressing ChR2 in D1-positive neurons in the striatum (*Drd1a::ChR2* mice). Mice had ad libitum access to food, but water was restricted. Free water was provided on days with no experimental sessions. All procedures were carried out in accordance

with National Institutes of Health Standards and were approved by the University of Oregon Institutional Animal Care and Use Committee.

**Study design and statistics.** Sample sizes (number of mice, behavioral sessions, and neurons) were based on previous literature in the field[10,29]. Two-sided non-parametric statistical tests were used with no assumption of normality of the sample distributions. When comparing laser stimulation in transgenic vs. wild-type mice, the experimenter was not blind to the genotype of animals, but data collection was automated to minimize potential biases.

**Behavioral task.** The two-alternative choice sound discrimination task was carried out inside single-walled sound-isolation boxes (IAC-Acoustics). Behavioral data was collected using the taskontrol platform (www.github.com/sjara/taskontrol) developed in our laboratory using the Python programming language (www.python.org). Mice initiated each trial by poking their noses into the center port of a three-port behavior chamber. After a silent delay of random duration (150–250 ms, uniformly distributed), a narrow-band sound (chord) was presented for 100 ms. Animals were required to stay in the center port until the end of the sound and then chose one of the two side ports for reward (2 µl of water) according to the frequency of the sound (low-frequency: left port; high-frequency: right port). If animals withdrew before the end of the stimulus, the trial was aborted and ignored in the analysis. Stimuli were chords composed of 12 simultaneous pure tones logarithmically spaced in the range $f/1.2$ to $1.2f$ for a given center frequency $f$. Within a behavioral session, we used 6 or 8 distinct center frequencies. The intensity of all sound components was set to the same value between 30–50 dB-SPL (changing from one trial to the next) during the initial training, but fixed during testing at 50 dB-SPL. Each behavioral session lasted 60–90 min.

Switching task: To test mice under changing stimulus-action associations, we used a variation of the two-alternative task described above[16]. A single session consisted of several blocks of 200–250 trials, with two (out of three) possible sound stimuli presented in each block. In a low-boundary block, mice were required to choose the left reward port after a low-frequency sound, and choose the right port for a middle-frequency sound. In a high-boundary block, mice were required to discriminate between the middle-frequency sound and a high-frequency sound, with the middle-frequency sound now being rewarded on the left port (Fig. 7a). No additional cue was given indicating the change in contingency. The initial contingency in a session was randomized from one day to the next.

**Cell counting.** To quantify the density of dMSNs along the anterior–posterior axis of the dorsal striatum, we used histology data from GENSAT on founder line FK150 (http://www.gensat.org/ShowMMRRCStock.jsp?mmrrc_id=MMRRC:036916). In situ hybridization was carried out with a probe against *Drd1a*. Four coronal sections from anterior striatum and four coronal sections from posterior striatum were selected, corresponding to our manipulation sites (0.5 mm anterior to Bregma and 1.7 mm posterior to Bregma, respectively). We counted the number of cells in eight non-overlapping 367.5 µm by 367.5 µm windows from each brain region and averaged the cell counts to yield the mean count for each region.

**Tetrode array implants.** Animals were anesthetized with isoflurane through a nose cone on the stereotaxic apparatus. Mice were surgically implanted with a custom-made microdrive containing eight tetrodes targeting the right posterior striatum. Each tetrode was composed of four tungsten wires (CFW0011845, California Fine Wire) twisted together. The eight tetrodes varied in length with 500 µm difference between the longest and the shortest tetrodes. Tetrodes were positioned at 1.7 mm posterior to bregma, 3.5–3.55 mm from midline, and 2 mm from the brain surface at the time of implantation. All animals were monitored and recovered fully before behavioral and electrophysiological experiments.

**Neural recordings.** Electrical signals were collected using an RHD2000 acquisition system (Intan Technologies) and OpenEphys software (www.open-ephys.org). Evoked responses to sound were monitored daily and tetrodes were moved down after each recording session. At the first depth where sound-evoked responses were observed, we started collecting electrophysiological data during the sound discrimination task. Recordings for each animal stopped when no more sound responses were observed. Electrodes were coated with DiI (Thermo Fisher Scientific, Cat #V22885) before implanting. Tetrode locations were confirmed histologically according to a standard brain atlas[31] based on electrolytic lesions and DiI fluorescence.

**Optogenetic stimulation in awake mice.** Optical fibers (CFML12U-20, ThorLabs) were cleaved and etched with hydrofluoric acid for 40 min to obtain a cone-shaped tip. Each optical fiber was glued to a metal guide tube that helped secure the fiber ferrule to the skull. Before implantation, optical fibers were connected to a blue laser (445 nm) built in-house and the light output calibrated using a PM100D power meter (ThorLabs). *Drd1a::ChR2* mice were randomly assigned to two groups and received optical fiber implant bilaterally targeting either the anterior dorsal striatum (DMS, 0.5 mm anterior to bregma, 1.6 mm from midline, and 2.5 mm

from brain surface) or the posterior striatum (1.7 mm posterior to bregma, 3.5 mm from midline, and 2.1 mm from brain surface). Optical fibers implanted in the posterior striatum were attached to a movable microdrive together with a tetrode bundle, such that only the depth of the optical fiber and tetrode bundle was varied between stimulation sessions. Fiber locations were verified histologically postmortem.

To assess the effect of stimulating striatal direct-pathway neurons on movement initiation, each mouse was placed in a square activity chamber (22 cm by 22 cm) and video-recorded from a camera mounted above the chamber. Two pieces of colored tape were placed on the implant to facilitate tracking. Optical stimulations were carried out unilaterally at 1 mW (measured at the tip of the fiber before implantation) for 1.5 s each time. Changes in the angle of the mouse's head were tracked across video frames (see Analysis of behavioral data for details on this analysis) using custom software written in Python (www.python.org) and OpenCV (opencv.org). To assess the effect of activating posterior striatal direct-pathway neurons during sound discrimination, unilateral optical stimulations were carried out while animals performed the two-alternative choice task. To assess the frequency preference of each stimulation site, neural recordings were made at the depth of the optical fiber via the attached tetrode bundle, while animals were presented with sound stimuli similar to those used in the task before each behavioral session. Each behavioral session lasted for 80–120 min during which animals performed 500–900 trials; laser stimulations were delivered in 20% of the trials. In a stimulation trial, the laser stimulation (1 mW, 200 ms) started 50 ms before the sound onset and ended 50 ms after the sound.

**Muscimol inactivation**. Bilateral craniotomies were performed under stereotactic surgery over the posterior striatum (1.7 mm posterior to bregma, 3.55 mm lateral from midline) of mice trained in the two-alternative choice sound discrimination task. Headbars were implanted to allow for head fixation. Each craniotomy was protected with a plastic ring and filled with silicon elastomer (Sylgard 170, Dow Corning). Animals were allowed to recover for at least 3 days before resuming behavioral training. Following recovery, implanted animals were trained on the sound discrimination task until they reached their pre-surgery performance level before beginning muscimol inactivation.

For intracranial injection, we used glass pipettes (5 μl Disposable Micropipettes, VWR) pulled and trimmed to an inner diameter of 15–20 mm at the tip. Animals were head-fixed and allowed to run on a wheel during the injection. Craniotomies were exposed by removing the silicon elastomer covering, and a glass pipette filled with reagent (either muscimol or saline) was lowered into the brain to a depth of 3.1 mm from brain surface using a micromanipulator. A volume of 45 nl of muscimol (0.25 mg ml$^{-1}$, final dose of 11.25 ng per hemisphere) was injected under air pressure in each hemisphere at a rate of 90 nl min$^{-1}$. Given the relationship between concentration and diffusion distance (from Fick's law) and previous reports of muscimol effects on neuronal activity[32], we expect that by the first 10 min of the behavioral session, the effects of muscimol (50% reduction in firing or more) will be confined to a volume smaller than 1 mm in diameter centered at the injection site. This volume matches well the extent of the posterior tail of the striatum (approx. 1 mm A-P, 0.6 mm M-L, 1.5 mm D-V) that receives auditory inputs[4]. The pipette was left in place for 60 s following the injection, then raised 0.5 mm and left in place for another 60 s before being removed. Injection in the second hemisphere was always completed within 10 min of the first injection. The craniotomies were then protected with a new silicon elastomer cap, and the mouse was placed back into its home cage for 30 min before starting the behavior session. After collection of four saline sessions and four muscimol sessions, 45 nl of fluorescent dye (DiI, Thermo Fisher Scientific) was injected at the same injection coordinates. Animals were perfused transcardially with 5% paraformaldehyde, and brains were extracted and postfixed for 12–24 h. Brains were then sliced (100 μm) and imaged to verify the location of fluorescent dye injection.

Fluorescent muscimol (Muscimol, BODIPY TMR-X Conjugate, Thermo Fisher Scientific) was dissolved in phosphate-buffered saline to a final concentration of 0.5 mg ml$^{-1}$. We followed the same protocol for intracranial injection as for muscimol. The injection volume was 360 nl per hemisphere (final dose of 180 ng per hemisphere) to account for the larger molecular weight and reduced spread of fluorescent muscimol. As with muscimol injections, animals rested for 30 min before starting the behavior session. Animals were euthanized and transcardially perfused with 5% paraformaldehyde within 1 h of finishing behavioral testing. Brains were extracted and postfixed overnight, and sliced (100 μm) to quantify spread of fluorescent muscimol.

**Analysis of behavioral data**. Estimation on head angle: Centroids for a green and red colored tape attached to the left and right side of each animal's head implant were estimated for each video frame. Head angle (Fig. 1b) was calculated from the line connecting these centroids. Head angle for frames in which colored tape was not visible was estimated using linear interpolation from other frames. Total change in head angle for each trial (in Fig. 1c) was calculated as the angle at the end of the stimulation minus the angle at the beginning of the stimulation. Inspection of the videos confirmed that body rotations occurred following head rotations larger than 90°.

Psychometric curve fitting for Figs. 2 and 3 was performed via constrained maximum likelihood to estimate the parameters of a logistic sigmoid function

(http://psignifit.sourceforge.net). To quantify rightward bias produced by optogenetic stimulation in each session, we calculated the difference between the fraction of rightward choices with stimulation and without stimulation (Fig. 2d).

**Analysis of neuronal data**. Data were analyzed using in-house software developed in Python (www.python.org). Spiking activity of single units was isolated using an automated expectation maximization algorithm (Klustakwik[33]). Isolated clusters were only included in the analysis if less than 2% of inter-spike intervals were shorter than 2 ms. Spike shapes were manually inspected to exclude noise signals.

A cell was considered sound-responsive if activity evoked by all sound frequencies pooled together (0–100 ms from sound onset) was significantly different from baseline spontaneous activity ($-100$ to 0 ms from sound onset) according to a Wilcoxon rank-sum test. The sound response index for each cell was calculated as $(S - B)/(S + B)$, where $S$ is the average sound-evoked response and $B$ is the average spontaneous firing rate. To determine whether neuronal firing encoded task-specific sound features, we calculated a selectivity index which quantified the difference between the firing rates in response to high-frequency vs. low-frequency sounds with respect to the categorization boundary: $(F_{high} - F_{low})/(F_{high} + F_{low})$.

For movement-related responses, average firing rates in a 50–150 ms window after the animal exited the center port were quantified. A movement modulation index (MI) was calculated by $(I - C)/(I + C)$, where $I$ and $C$ were average firing rates from trials with movement to the reward port ipsilateral and contralateral to the recording site, respectively. To ask whether neuronal activity encoded movement direction, firing rates for trials with contralateral vs. ipsilateral movement were compared using Wilcoxon rank-sum test; neurons with a resulting $p$-value of less than 0.05 were considered selective to movement direction.

To evaluate whether neuronal response predicted the animal's choice, we focused on neurons that showed sound-evoked response to the stimuli with ambiguous behavioral choices (stimuli near discrimination boundary in the sound discrimination task) or periodically updated categorization contingencies (intermediate frequency stimulus in the switching task). To assess responsiveness to the stimuli of interest, spike counts were quantified in non-overlapping bins of 25 ms during the response period (0–100 ms after sound onset) and during the baseline period (25–50 ms before sound onset). A test statistic ($z$-score) was computed for each response bin in relation to the baseline bin using a Wilcoxon rank-sum test. We considered a cell responsive if the $z$-score of any bin during the response period fell outside the range ($-3$, 3). To quantify the impact of choice on neuronal activity, we calculated a MI using the equation: $(I - C)/(I + C)$, where $I$ and $C$ were average firing rate in trials with ipsilateral and contralateral choices, respectively. Only sessions with at least 60% correct trials were included in the switching task analysis. To ensure that the MI reflected neuronal activity during stable performance of the task, the first 20 trials after a contingency switch were excluded and only correct trials were included in the MI calculation. We tested statistical significance of choice modulation for each cell using the Wilcoxon rank-sum test between the evoked firing of each choice (sound discrimination task) or contingency (switching task). In the sound discrimination task, this analysis focused on the sound stimulus that elicited the largest sound-evoked response out of the two most ambiguous stimuli closest to the categorization boundary. To exclude effects of nonstationarity fluctuations in neuronal firing rate over time in the switching task, cells were counted as significantly modulated only if the modulation effect was observed in at least two different switches of contingency blocks.

**Data availability**. Behavioral data was collected using the taskontrol platform developed in our laboratory (http://www.github.com/sjara/taskontrol). Psychometric curve fitting was performed using the psignifit package for Python (http://psignifit.sourceforge.net). The data that support the findings of the current study are available from the corresponding author upon reasonable request.

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

## Acknowledgements

The authors thank members of the Jaramillo Lab for discussion and comments on the manuscript. This research was supported by the National Institute on Deafness and Other Communication Disorders (R01DC015531), the Medical Research Foundation, and the Office of the Vice President for Research & Innovation at the University of Oregon.

## Author contributions

S.J. conceived the project. S.J., W.I.W., L.G., and N.D.P. designed the experiments. W.I.W., P.L.P., and L.G. conducted and analyzed the electrophysiological recordings. L.G. conducted and analyzed the optogenetic studies. N.D.P. conducted and analyzed the muscimol inactivation studies. S.J. supervised all aspects of the work. L.G., N.D.P., and S.J. wrote the paper.

## Additional information

**Competing interests:** The authors declare no competing interests.

