## [Peer Review File · Nature Communications]

Reviewers' comments:

Reviewer #1 (Remarks to the Author):

Jaramillo lab presents physiological and direct-manipulation data that supports the role of the striatum in auditory decision making. The paper is carefully conducted and the experiments are well thought through.

My concerns and suggestions are related to the interpretation of the data.

The first part of the experiment states that posterior striatal neurons do "not directly drive movement". This is very important for the subsequent claims.

However, currently, most influential ideas of the basal ganglia (BG) suggest that BG acts as a gate not a "driver". For example, stimulation of the primate caudate nucleus may not necessarily drive saccades in a stim-evoked manner (e.g. if we stimulate, the movement will not occur "each time you stim at the same time relative to the stim"), however, the probability of movement to the contralateral field will be increased for some time (see Yamamoto et al. Journal of Neuroscience). Is there any analyses the authors can do to convince us that these neurons are not gating movements (in a sensory-related manner)? Do the authors have any data in which animals are simply performing an instrumental task (no decision, and just move left or right, for example) that would show that stimulation does not affect reaction times or likelihood of movement? If these data can not be provided, it may be important to weaken the claims that the neurons are not action-control related. A neuron can have "sensory" activity, as the authors nicely show, but can still effectively work to bias action (when the appropriate sensory stimulus is delivered). I think its very important to consider that more carefully.

What is the quantitative assessment for the claim that neurons contained more info about sound than choice? I recommend using a more quantitative approach to decode the information in the neuronal activity to strengthen these claims.

Reviewer #2 (Remarks to the Author):

Main Review:

Guo and colleagues train mice on an auditory frequency discrimination task to determine what role a region of the striatum (posterior tail of the dorsal striatum, referred to throughout as the posterior striatum) plays in auditory-guided decisions. The authors find that activation of direct pathway posterior striatum neurons (e.g. D1R-expressing MSNs) does not drive rotational head movements outside of task engagement, unlike stimulation of other striatal regions. In contrast, stimulating these same neurons during an auditory task does bias decisions toward the direction contralateral to the stimulation hemisphere. Pharmacologically bilateral silencing of posterior striatum (non-specifically) causes performance on all sound frequencies to drop to near chance. Electrophysiological recordings from posterior striatum (non-specific) show that many neurons are responsive to

sound and to movement direction, but very few are responsive to choice. These findings are further recapitulated in a task in which learning of a new frequency boundary occurs online during the recordings. From these experiments, the authors conclude that neural activity in the posterior striatum reflects a stable representation of stimulus identify (i.e. sound frequency) and sometimes movement direction, but not choice.

From my reading, this is the first deep dive into the role of the posterior striatum (yet is unclear exactly how this region differs from that studied by Zador and colleagues in two previous papers [see my comments below beginning with "Line 26-39]). The finding that activation of D1R expressing posterior striatal neurons biases choice, yet they do not bias movement, is interesting and suggest a gate that must be open for these neurons to influence behavior. It is then surprising, however, that the activity of posterior striatal neurons does not encode choice, and that, as far as I can tell, high vs. low frequencies are not differentially encoded in different hemispheres; these findings seem a bit incongruous. The results seem largely similar to what has been observed in auditory cortex (Znamenskiy and Zador). My primary interpretation is that, aside from potentially being a relay station, the posterior striatum serves a role in auditory-guided decisions not so different than primary auditory cortex.

Major Concerns:

1. The authors state that optogenetic stimulation of D1R-expressing MSN's in the posterior striatum does not drive movement (line 78). This is a broad statement, yet the only movement quantified is head movements. Behavioral responses used by the mice to indicate a decision likely also involve directional body movements (e.g. although I realize the cartoons in Fig. 2A are illustrative, they indeed show mice moving their whole bodies without any head rotation). The authors should state/show whether any movements other than head rotation —spontaneous or changes to ongoing movements —were detected following posterior striatum stimulation. Alternatively, the authors should show that behavioral responses during the task use only rotational head movements. In general, a more thorough quantification of the motor-related effects of posterior striatum stimulation (particularly a lack thereof) will make even more interesting the authors findings of effects of stimulation during task engagement.

2. Line 144 - Asking whether neurons differentially respond to high vs. low frequencies as a way of determining whether they convey task-relevant information here seems a bit misleading. Is the activity reflecting a categorical boundary (as suggested by this paragraph)? Or are these neurons frequency selective from the get-go (inherited from AC and MGB), and therefore some fraction will be task relevant so long as the task requires distinguishing sounds along the stimulus dimension defined by frequency? In particular I would like to know the following: Of the 137 cells that provided task-relevant information: did the change in their firing rate occur at the discrimination boundary defined by the experimenters? Of the 95 cells that were sound responsive but did not provide task relevant information, were they particularly tuned to frequencies at the boundary? Or just broadly tuned?

3. I am surprised that the distribution of posterior stratum neurons responsive to movement

in the ipsi and contra directions are largely equivalent (55% vs. 45%) given the strong behavioral ramifications of stimulating this region during task engagement. One major difference between the stimulation and recording experiments is the targeting of specific cell types (i.e. D1R expressing cells during stimulation; blind recordings during physiology). I do not believe it is necessary for the authors to re-do these experiments making recordings from photo-identified D1R-expressing neurons (yet I'd welcome it if they did, and believe it would add substantially to their story). At the least, the authors should address this discrepancy in the discussion.

4. By analyzing neural activity during trials with sounds at the discrimination boundary, the authors find no (or very little) choice activity in posterior striatum neurons. They state (on line 197) that these results do not rule out choice activity when the animal has formed a clear stimulus-action association. Yet, if my interpretation is correct, the authors have the data to test this by looking at easy trials with high or low frequency sounds (i.e.. those with clear stimulus-action associations). First, are the high frequency neurons located in one hemisphere and the low frequency neurons in the other? Second, of the 62 neurons (line 164) that are responsive to sounds and movements, is there a correlation between high vs. low frequency tuning and left vs. right choice activity?

5. For the data presented in Fig. 7, it would be informative and more conclusive to plot sound-evoked firing rate as a function of trial number, for several trials preceding and several trials following a transition boundary. Moreover, how quickly does the behavioral learning take place? Does it take place over just a couple of trials, and do the mice even learn at all before the next transition? Some behavioral data should be shown here to indicate that the mice are learning and not just confused.

6. The word "flexible" in the title does not seem to be warranted, given that only the data in Fig. 7 use a task in which there is a flexible decision boundary.

Minor comments and concerns:

General - I had a hard time following the nomenclature at the beginning of the results section. Several of the following comments are suggestions that the authors may take to make this manuscript more readable for future readers.

Line 26-39 - It would be helpful to have, perhaps as a supplement, a figure panel illustrating a schematic of the striatum and the various regions defined classically (i.e. DMS and DLS) and more recently (e.g. posterior tail of dorsal striatum; anterior dorsal striatum). For a reader who does not study the striatum but for whom this work is nonetheless extremely interesting, it can be hard to remember whether these are all mutually exclusive regions, whether one is a subregion of the other, or whether the new and old maps are completely out of sync with each other.

Line 40 - add a parenthetical to stay that "...posterior tail of the dorsal striatum (referred to from here on as posterior striatum) in rodents receives...".

Line 66 and 70 - are "anterior dorsal striatal" neurons (line 66) the same as neurons in the "anterior striatum" (line 70)? Please keep a standard nomenclature throughout and, as in previous comment, let the reader know if you will be abbreviating going forward.

Line 74 - Is "dorsomedial striatum" the same as "anterior dorsal" and "anterior" striatum mentioned earlier in paragraph? Context clues suggest this to be the case, but unclear.

Line 106 - Clarification on how the location and tuning experiments were performed (Supp 1 and Supp 2). Were these posthoc analyses done in retrospect using the fixed positioning of a fiber in each experiment? Or was the fiber + tetrode bundle systematically moved around to test these two parameters?

Line 131 - "but not for executing the movements required by the task...". Again, unclear from the data presented that a head rotation is the movement required for the task.

Line 282 - should be "sound-driven decision"

Fig. 2D - Show data as in Fig 1C, with data points segregated by mouse and means for each mouse as filled circles.

Fig. 4-7 - It would be helpful if these rasters also showed the time of movement onset for each trial (fig. 4) and the time of sound onset for each trial (fig. 5-7), perhaps with a colored dot on each trial at the appropriate time.

Reviewer #3 (Remarks to the Author):

This is an interesting, well-written manuscript. The authors carried out experiments to stimulate, suppress and record neural activity in the striatum of behaving mice and found that the posterior dorsal striatum is required to perform a frequency discrimination task but exhibits unexpectedly stable representations of the acoustic stimuli that the mice were presented with; unexpectedly, in the sense that the dorsal posterior striatum may be expected to drive behavioural choices. However, animals' behavioural choice modulated sound evoked responses in only a very small minority (no larger than the number of choice-modulated neurons in the auditory cortex and thalamus of rats) of striatal neurons. I have some concerns but consider this manuscript potentially suitable for publication provided the concerns can be addressed.

Line 68: Optogenetic stimulation: How are the ChR2-expressing dMSNs distributed along the anterior-posterior axis. Could it simply be that fewer neurons are activated during optogenetic stimulation of the posterior striatum and that, therefore, we do not see any stimulation-evoked head or body rotation? Please show some histology and/or find a way to quantify the number of neurons likely activated in each condition.

86: Only mice with optogenetic implants in the posterior striatum were tested in the behavioural task. Why not also the ones with the implant in the anterior striatum? It would

be interesting to see what happens when these mice are stimulated during behaviour. Perhaps the data already exist?

99: What is the distribution of dMSNs along the mediolateral axis. Could it be that fewer neurons are activated with the implant in the border region between cortex and striatum and that that is responsible for the difference in bias?

Figure S2: Are the plotted preferred frequencies the average of all units recorded at a given site?

122: What is the rationale for using muscimol for the inactivation experiments rather than an optogenetic approach as with the stimulation experiments?

130: "These inactivation results indicate that the activity of neurons in the posterior striatum is necessary for auditory decisions, but not for executing the movements required by the task."

Some of the results (fewer trials, slower withdrawals from center spout) indicate the opposite to what is said in the underlined part of this sentence.

142: typo: neurons'

148: typo: characterized

179: "...and the number of left and right choices was about the same." Please be more specific.

185: Please provide rationale behind spike shape analysis.

267: "This result is comparable to observations from the auditory thalamus and auditory cortex..." Do we know about the anterior striatum?

320: typo: result

325: Why only male mice?

384: "Before implant" should say 'Before implantation'.

Muscimol/Saline injections: One might expect brain damage from repeated injections: Did the animals' performance deteriorate as a function of the number of injections? How far did the muscimol spread? Mediolateral as well as rostrocaudal. The injection volume of the fluorescent muscimol was quite large (360nl per hemisphere). Could that have caused brain damage? Rostrocaudal spread?

Please find below our point-by-point response to the three reviewers' comments. We have highlighted all the changes in the manuscript text file in red, as well as provided the corresponding line numbers of each edit in the response.

Reviewer #1 (Remarks to the Author):

Jaramillo lab presents physiological and direct-manipulation data that supports the role of the striatum in auditory decision making. The paper is carefully conducted and the experiments are well thought through.

My concerns and suggestions are related to the interpretation of the data.

The first part of the experiment states that posterior striatal neurons do "not directly drive movement". This is very important for the subsequent claims.

However, currently, most influential ideas of the basal ganglia (BG) suggest that BG acts as a gate not a "driver". For example, stimulation of the primate caudate nucleus may not necessarily drive saccades in a stim-evoked manner (e.g. if we stimulate, the movement will not occur "each time you stim at the same time relative to the stim"), however, the probability of movement to the contralateral field will be increased for some time (see Yamamoto et al. Journal of Neuroscience). Is there any analyses the authors can do to convince us that these neurons are not gating movements (in a sensory-related manner)? Do the authors have any data in which animals are simply performing an instrumental task (no decision, and just move left or right, for example) that would show that stimulation does not affect reaction times or likelihood of movement? If these data can not be provided, it may be important to weaken the claims that the neurons are not action-control related. A neuron can have "sensory" activity, as the authors nicely show, but can still effectively work to bias action (when the appropriate sensory stimulus is delivered). I think its very important to consider that more carefully.

Response: *We have modified the Results (lines 82-83) to avoid describing the striatum as a "driver" of actions. Although we don't have data of animals performing an instrumental task, we have added a paragraph to the Discussion (lines 276-278) describing how our data is consistent with posterior striatal neurons biasing actions during sound-driven decisions when the appropriate sensory stimulus is present.*

What is the quantitative assessment for the claim that neurons contained more info about sound than choice? I recommend using a more quantitative approach to decode the information in the neuronal activity to strengthen these claims.

Response: *We have now calculated the ability of each neuron to discriminate task-relevant sounds and to discriminate choice before action initiation, and report a comparison of the fraction of cells able to differentiate different sounds versus different choices (lines 211-214).*

We also modified the Discussion to clarify that neurons in this region display sound representations that were minimally influenced by choice (lines 259-262).

Reviewer #2 (Remarks to the Author):

Main Review:

Guo and colleagues train mice on an auditory frequency discrimination task to determine what role a region of the striatum (posterior tail of the dorsal striatum, referred to throughout as the posterior striatum) plays in auditory-guided decisions. The authors find that activation of direct pathway posterior striatum neurons (e.g. D1R-expressing MSNs) does not drive rotational head movements outside of task engagement, unlike stimulation of other striatal regions. In contrast, stimulating these same neurons during an auditory task does bias decisions toward the direction contralateral to the stimulation hemisphere. Pharmacologically bilateral silencing of posterior striatum (non-specifically) causes performance on all sound frequencies to drop to near chance. Electrophysiological recordings from posterior striatum (non-specific) show that many neurons are responsive to sound and to movement direction, but very few are responsive to choice. These findings are further recapitulated in a task in which learning of a new frequency boundary occurs online during the recordings. From these experiments, the authors conclude that neural activity in the posterior striatum reflects a stable representation of stimulus identity (i.e. sound frequency) and sometimes movement direction, but not choice.

From my reading, this is the first deep dive into the role of the posterior striatum (yet is unclear exactly how this region differs from that studied by Zador and colleagues in two previous papers [see my comments below beginning with “Line 26-39]). The finding that activation of D1R expressing posterior striatal neurons biases choice, yet they do not bias movement, is interesting and suggest a gate that must be open for these neurons to influence behavior. It is then surprising, however, that the activity of posterior striatal neurons does not encode choice, and that, as far as I can tell, high vs. low frequencies are not differentially encoded in different hemispheres; these findings seem a bit incongruous. The results seem largely similar to what has been observed in auditory cortex (Znamenskiy and Zador). My primary interpretation is that, aside from potentially being a relay station, the posterior striatum serves a role in auditory-guided decisions not so different than primary auditory cortex.

Major Concerns:

1. The authors state that optogenetic stimulation of D1R-expressing MSN's in the posterior striatum does not drive movement (line 78). This is a broad statement, yet the only movement quantified is head movements. Behavioral responses used by the mice to indicate a decision likely also involve directional body movements (e.g. although I realize the cartoons in Fig. 2A are illustrative, they indeed show mice moving their whole bodies without any head rotation). The authors should state/show whether any movements other than head rotation—spontaneous or changes to ongoing movements—were detected following posterior striatum stimulation. Alternatively, the authors should show that behavioral responses during the task

use only rotational head movements. In general, a more thorough quantification of the motor-related effects of posterior striatum stimulation (particularly a lack thereof) will make even more interesting the authors findings of effects of stimulation during task engagement.

Response: *We have modified the text in Results and Methods to report that body rotation is observed following anterior striatum stimulation but not posterior striatum stimulation (lines 67-68, lines 482-483). The reviewer is correct in noting that our task requires directional head and body movements from the mice.*

2. Line 144 - Asking whether neurons differentially respond to high vs. low frequencies as a way of determining whether they convey task-relevant information here seems a bit misleading. Is the activity reflecting a categorical boundary (as suggested by this paragraph)? Or are these neurons frequency selective from the get-go (inherited from AC and MGB), and therefore some fraction will be task relevant so long as the task requires distinguishing sounds along the stimulus dimension defined by frequency? In particular I would like to know the following: Of the 137 cells that provided task-relevant information: did the change in their firing rate occur at the discrimination boundary defined by the experimenters? Of the 95 cells that were sound responsive but did not provide task relevant information, were they particularly tuned to frequencies at the boundary? Or just broadly tuned?

Response: *We have modified the text to indicate that the posterior striatal neurons display tuning across the frequency spectrum, and because of this they are expected to provide task relevant information (lines 154-157). Most of the sound responsive cells that did not differentiate between high vs. low frequency were broadly tuned.*

Although in this study we do not report sound response properties from posterior striatal neurons in naive mice, Supplementary Figure 6 shows that these neurons display frequency selectivity outside the behavioral task.

In addition, results from the switching task (Figure 7) indicate that changes in categorization boundary did not influence the responses of the cells.

3. I am surprised that the distribution of posterior stratum neurons responsive to movement in the ipsi and contra directions are largely equivalent (55% vs. 45%) given the strong behavioral ramifications of stimulating this region during task engagement. One major difference between the stimulation and recording experiments is the targeting of specific cell types (i.e. D1R expressing cells during stimulation; blind recordings during physiology). I do not believe it is necessary for the authors to re-do these experiments making recordings from photo-identified D1R-expressing neurons (yet I'd welcome it if they did, and believe it would add substantially to their story). At the least, the authors should address this discrepancy in the discussion.

Response: *The manuscript has been updated to clarify that the physiology recordings likely include both direct- and indirect-pathway MSNs, while the stimulation only activated dMSNs (lines 169-172). Therefore the largely equivalent proportions of neurons responsive to ipsi versus contra movement could be a result of recording from unspecified cell types.*

4. By analyzing neural activity during trials with sounds at the discrimination boundary, the authors find no (or very little) choice activity in posterior striatum neurons. They state (on line 197) that these results do not rule out choice activity when the animal has formed a clear stimulus-action association. Yet, if my interpretation is correct, the authors have the data to test this by looking at easy trials with high or low frequency sounds (i.e., those with clear stimulus-action associations). First, are the high frequency neurons located in one hemisphere and the low frequency neurons in the other? Second, of the 62 neurons (line 164) that are responsive to sounds and movements, is there a correlation between high vs. low frequency tuning and left vs. right choice activity?

Response: *While the ‘easy trials’ indeed have stable stimulus-action associations, for such ‘easy’ stimuli our mice usually performed well above 80% accuracy. Therefore, within a single session, we get very few numbers of trials for one of the two possible choices for an ‘easy’ stimulus. This makes it challenging to make a meaningful statistical comparison in these cases.*

Regarding the frequency selectivity in each hemisphere, we observed neurons that preferred either the ‘high’ or ‘low’ frequency sounds in the right hemisphere (Figure 4D). In addition, in our optogenetic stimulation experiments, we recorded multiunit sound responses from both hemispheres and observed that both hemispheres displayed a comparable range of preferred frequencies (Supp Figure 2A).

We now included (lines 176-180) an analysis of correlation between high vs. low frequency selectivity and left vs. right movement selectivity, and found that they are positively correlated.

5. For the data presented in Fig. 7, it would be informative and more conclusive to plot sound-evoked firing rate as a function of trial number, for several trials preceding and several trials following a transition boundary. Moreover, how quickly does the behavioral learning take place? Does it take place over just a couple of trials, and do the mice even learn at all before the next transition? Some behavioral data should be shown here to indicate that the mice are learning and not just confused.

Response: *We added a supplementary figure to illustrate how fast the animals switch between contingencies (Supp. Figure 11, lines 237-238), and added a panel to Figure 7 showing the behavioral dynamics of switching (Figure 7B). Given that animals switch in a few trials, there is not enough data to evaluate the neuronal response during contingency switch under these conditions. Therefore, we focused our analysis on the neuronal representation of sound and choice after a new contingency has been established.*

6. The word “flexible” in the title does not seem to be warranted, given that only the data in Fig. 7 use a task in which there is a flexible decision boundary.

Response: *Although the reviewer is correct in noting that only the results from Figure 7 use a task in which there is a flexible decision boundary, we would argue that all other experiments in the paper provide the context for testing the role of posterior striatal neurons in flexible auditory decisions. This is an important part of the message of our paper which we believe warrants a mention in the title.*

Minor comments and concerns:

General - I had a hard time following the nomenclature at the beginning of the results section. Several of the following comments are suggestions that the authors may take to make this manuscript more readable for future readers.

Line 26-39 - It would be helpful to have, perhaps as a supplement, a figure panel illustrating a schematic of the striatum and the various regions defined classically (i.e. DMS and DLS) and more recently (e.g. posterior tail of dorsal striatum; anterior dorsal striatum). For a reader who does not study the striatum but for whom this work is nonetheless extremely interesting, it can be hard to remember whether these are all mutually exclusive regions, whether one is a subregion of the other, or whether the new and old maps are completely out of sync with each other.

Line 40 - add a parenthetical to stay that "...posterior tail of the dorsal striatum (referred to from here on as posterior striatum) in rodents receives...".

Line 66 and 70 - are "anterior dorsal striatal" neurons (line 66) the same as neurons in the "anterior striatum" (line 70)? Please keep a standard nomenclature throughout and, as in previous comment, let the reader know if you will be abbreviating going forward.

Line 74 - Is "dorsomedial striatum" the same as "anterior dorsal" and "anterior" striatum mentioned earlier in paragraph? Context clues suggest this to be the case, but unclear.

Response: *We have fixed the nomenclature used throughout the paper as the reviewer suggested. We now clarify the relation between 'anterior dorsal striatum' and the classically defined DMS and DLS regions (lines 35-36). We indicate that 'anterior striatum' is used throughout to refer to 'anterior dorsal striatum' (line 62-63). And 'posterior striatum' is used throughout to refer to 'posterior tail of striatum', as stated on line 40).*

Line 106 - Clarification on how the location and tuning experiments were performed (Supp 1 and Supp 2). Were these posthoc analyses done in retrospect using the fixed positioning of a fiber in each experiment? Or was the fiber + tetrode bundle systematically moved around to test these two parameters?

Response: *We have modified the methods to clarify the experimental procedure (lines 422). Briefly, the medial-lateral location of the fiber + tetrode bundle were fixed after implantation, and we manipulated the depth of the tip of the whole bundle (using a movable drive) in different sessions to record from different sites in the posterior striatum.*

Line 131 - “but not for executing the movements required by the task...”. Again, unclear from the data presented that a head rotation is the movement required for the task.

Response: *We have removed this sentence and report the motor impairments observed during muscimol experiments (line 130).*

Line 282 - should be “sound-driven decision”
Fixed.

Fig. 2D - Show data as in Fig 1C, with data points segregated by mouse and means for each mouse as filled circles.

Response: *We have updated Figure 2D (and corresponding caption) to segregate the data points by mouse and display the mean bias for each mouse.*

Fig. 4-7 - It would be helpful if these rasters also showed the time of movement onset for each trial (fig. 4) and the time of sound onset for each trial (fig. 5-7), perhaps with a colored dot on each trial at the appropriate time.

Response: *We now added box plots on top of rasters in Figures 4-7 to illustrate the distribution of sound onset or movement onset times for each session. We opted for box plots since our attempt at plotting colored dots on each trial obscured the main data.*

Reviewer #3 (Remarks to the Author):

This is an interesting, well-written manuscript. The authors carried out experiments to stimulate, suppress and record neural activity in the striatum of behaving mice and found that the posterior dorsal striatum is required to perform a frequency discrimination task but exhibits unexpectedly stable representations of the acoustic stimuli that the mice were presented with; unexpectedly, in the sense that the dorsal posterior striatum may be expected to drive behavioural choices. However, animals' behavioural choice modulated sound evoked responses in only a very small minority (no larger than the number of choice-modulated neurons in the auditory cortex and thalamus of rats) of striatal neurons. I have some concerns but consider this manuscript potentially suitable for publication provided the concerns can be addressed.

Line 68: Optogenetic stimulation: How are the ChR2-expressing dMSNs distributed along the anterior-posterior axis. Could it simply be that fewer neurons are activated during optogenetic stimulation of the posterior striatum and that, therefore, we do not see any stimulation-evoked head or body rotation? Please show some histology and/or find a way to quantify the number of neurons likely activated in each condition.

Response: *We have added a quantification of the number of dMSNs along the anterior-posterior axis, and indicated that we found a similar density of cells in the two regions stimulated (lines 76-81).*

86: Only mice with optogenetic implants in the posterior striatum were tested in the behavioural task. Why not also the ones with the implant in the anterior striatum? It would be interesting to see what happens when these mice are stimulated during behaviour. Perhaps the data already exist?

Response: *Although the anterior striatum was not the main focus of our study, we did test changes in performance during optogenetic stimulation in the anterior striatum in the sound discrimination task in one mouse. A contralateral choice bias was observed in trials with unilateral stimulation compared to control trials (18% for right hemisphere stimulation, 25% for left hemisphere stimulation, 2 sessions each hemisphere). We did not think that it was sufficient data to report in the manuscript.*

99: What is the distribution of dMSNs along the mediolateral axis. Could it be that fewer neurons are activated with the implant in the border region between cortex and striatum and that that is responsible for the difference in bias?

Response: *We believe the reviewer is correct in that fewer dMSNs were likely activated with the implant in the border region, as demonstrated in Supplementary Figure 1B, but this likely due to less laser light reaching the striatum, rather than a change in the density of dMSNs (confirmed from histological sections).*

Figure S2: Are the plotted preferred frequencies the average of all units recorded at a given site?

Response: *We now state in the Results and in the figure legend that the data in Supplementary Figure 2 correspond to the preferred frequency of the multiunit activity at a given site (lines 105-106).*

122: What is the rationale for using muscimol for the inactivation experiments rather than an optogenetic approach as with the stimulation experiments?

Response: *Although optogenetic approaches can provide higher temporal precision, we had not validated optogenetic silencing methods in our laboratory (for example characterizing possible rebound firing and the duration cells can be effectively suppressed by light). In contrast, we had validated muscimol inactivation at the time this study was performed.*

130: "These inactivation results indicate that the activity of neurons in the posterior striatum is necessary for auditory decisions, but not for executing the movements required by the task." Some of the results (fewer trials, slower withdrawals from center spout) indicate the opposite to what is said in the underlined part of this sentence.

Response: We have modified the result text (line 130) to clarify the results on motor impairment during muscimol experiments.

142: typo: neurons'

Fixed.

148: typo: characterized

Fixed.

179: "...and the number of left and right choices was about the same." Please be more specific.

Fixed.

185: Please provide rationale behind spike shape analysis.

Response: we now added the rationale for spike shape analysis in the text (lines 200-201).

267: "This result is comparable to observations from the auditory thalamus and auditory cortex..." Do we know about the anterior striatum?

Response: We do not have data from the anterior striatum to answer this question.

320: typo: result

Fixed.

325: Why only male mice?

Response: We used only male mice in this study to minimize behavioral variability. In addition, male mice are usually bigger such that head implants have a smaller effect on behavioral performance.

384: "Before implant" should say "Before implantation".

Fixed.

Muscimol/Saline injections: One might expect brain damage from repeated injections: Did the animals' performance deteriorate as a function of the number of injections?

Response: We have modified the Results to indicate that the performance for all mice tested was stable and did not deteriorate with repeated injections of saline (lines 123-124).

How far did the muscimol spread? Mediolateral as well as rostrocaudal.

Response: *We have updated the Methods to indicate the expected spread of muscimol (lines 453-458).*

The injection volume of the fluorescent muscimol was quite large (360nl per hemisphere). Could that have caused brain damage? Rostrocaudal spread?

Response: *We now report the rostrocaudal spread of fluorescent muscimol in the legend of Supplementary Figure 4. We did not observe impairment of task performance after injection of saline at 360 nl per hemisphere, suggesting absence of brain damage with this amount of fluid injected into the posterior striatum.*

REVIEWERS' COMMENTS:

Reviewer #1 (Remarks to the Author):

My major concerns have been addressed.

I have a few additional comments.

1) the authors need to consider previous work on perceptual decision bias and striatum (Wang et al., 2018, Neuron)

2) authors need need to consider a huge body of work on posterior striatum in monkeys starting with Work of Brown and Desimone, Yamamoto et al, and Kim Hyoung and Hikosaka - which particularly concerns itself with stability and processing of higher order visual information.

This treatment will make your claims stronger and relatevthem better to human circuitry.

Reviewer #2 (Remarks to the Author):

I believe this in an important contribution to our understanding of how the striatum contributes to auditory-guided behaviors, and that it is a nicely executed study.

Guo, Jaramillo, and colleagues have adequately addressed all of the concerns that I brought up in my initial review.

Although I still have some reservation about the term "flexible" in the title, I understand why the authors prefer to use this terminology, and I defer to them.

Reviewer #3 (Remarks to the Author):

My concerns have been addressed adequately and I consider the manuscript now suitable for publication.

Line 79: 'number' should say 'numbers'.

Response to reviewers

Reviewer #1 (Remarks to the Author):

My major concerns have been addressed.

I have a few additional comments.

1) the authors need to consider previous work on perceptual decision bias and striatum (Wang et al., 2018, Neuron)

RESPONSE: text has been added to the Discussion citing previous work on perceptual bias caused by striatal activation.

2) authors need need to consider a huge body of work on posterior striatum in monkeys starting with Work of Brown and Desimone, Yamamoto et al, and Kim Hyung and Hikosaka - which particularly concerns itself with stability and processing of higher order visual information. This treatment will make your claims stronger and relate them better to human circuitry.

RESPONSE: text has been added to the Introduction describing the role of primate caudate neurons in visual perceptual decision tasks.

Reviewer #2 (Remarks to the Author):

I believe this in an important contribution to our understanding of how the striatum contributes to auditory-guided behaviors, and that it is a nicely executed study.

Guo, Jaramillo, and colleagues have adequately addressed all of the concerns that I brought up in my initial review.

Although I still have some reservation about the term "flexible" in the title, I understand why the authors prefer to use this terminology, and I defer to them.

RESPONSE: No additional comments to address.

Reviewer #3 (Remarks to the Author):

My concerns have been addressed adequately and I consider the manuscript now suitable for publication.

Line 79: 'number' should say 'numbers'.

RESPONSE: The typo has been fixed. No additional comments to address.